# Suicide-Gene-Modified Extracellular Vesicles of Human Primary Uveal Melanoma in Future Therapies

**DOI:** 10.3390/ijms241612957

**Published:** 2023-08-19

**Authors:** Jana Jakubechova, Bozena Smolkova, Alena Furdova, Lucia Demkova, Ursula Altanerova, Andreas Nicodemou, Tatiana Zeleznikova, Daniela Klimova, Cestmir Altaner

**Affiliations:** 1Cancer Research Institute, Biomedical Center, Slovak Academy of Sciences, 845 05 Bratislava, Slovakia; 2Stem Cell Preparation Department, St. Elisabeth Cancer Institute, 812 50 Bratislava, Slovakia; ursula.altanerova@ousa.sk (U.A.);; 3Department of Ophthalmology, Faculty of Medicine, Comenius University, 814 99 Bratislava, Slovakia; 4Institute of Medical Biology, Genetics and Clinical Genetics, Faculty of Medicine, Comenius University, 814 99 Bratislava, Slovakia; andreas.nicodemou@fmed.uniba.sk (A.N.);

**Keywords:** primary uveal melanoma, transduction with suicide gene, therapeutic extracellular vesicles, gene-directed enzyme prodrug therapy

## Abstract

Extracellular vesicles secreted from uveal melanoma (UM) cells are involved in the establishment of the premetastatic niche and display transforming potential for the formation of metastases, preferentially in the liver. In this study, we cultivated human primary UM cells and uveal melanoma-associated fibroblasts in vitro to be transduced by infection with a retrovirus containing the suicide gene—fused yeast cytosine deaminase::uracil phospho-ribosyl transferase (*yCD::UPRT*). A homogenous population of *yCD::UPRT*-UM cells with the integrated provirus expressed the gene, and we found it to continuously secrete small extracellular vesicles (sEVs) possessing mRNA of the suicide gene. The *yCD::UPRT*-UM-sEVs were internalized by tumor cells to the intracellular conversion of the prodrug 5-fluorocytosine (5-FC) to the cytotoxic drug 5-fluorouracil (5-FU). The host range of the *yCD::UPRT*-UM-sEVs was not limited to UMs only. The *yCD::UPRT*-UM-sEVs inhibited the growth of the human cutaneous melanoma cell line A375 and uveal melanoma cell line MP38, as well as other primary UMs, to various extents in vitro. The *yCD::UPRT*-UM-sEVs hold the therapeutic and prophylactic potential to become a therapeutic drug for UM. However, the use of *yCD::UPRT*-UM-sEVs must first be tested in animal preclinical studies.

## 1. Introduction

Uveal melanoma is a rare malignant intraocular tumor occurring mainly in adults, representing approximately 5% of the total incidence of adult melanomas. Most UM arises from melanocytes within the choroid and, in contrast to cutaneous melanomas, is characterized by a low burden of genetic alterations. Despite originating from the same embryonic origin, uveal and cutaneous melanomas differ in their molecular characteristics and how they metastasize [1,2,3]. Primarily, skin-localized melanoma metastasizes in various organs, while UM is characterized by a high frequency of metastases in the liver, which are particularly resistant to treatment [4]. Primary UM can be effectively treated with enucleation, followed by stereotactic radiotherapy. While a few patients have detectable metastases at the time of diagnosis, about half of all patients will develop liver metastases within 5 years [5]. 

Up to now, no systemic therapy for patients with metastatic uveal melanoma has received regulatory approval. These patients have a dismal prognosis because of the limited efficacy of the available treatments [6]. The current chemotherapeutic approaches for UM metastases tested in clinical trials yield very low response rates [7,8]. A limited therapeutic effect was reported in an immunological clinical study using the adoptive transfer of tumor-infiltrating lymphocytes [9]. These therapeutic failures highlight the need for more targeted therapy that would specifically attack or prevent UM metastases [10]. 

Tumor cells with the potential to metastasize and be released into circulation retain their original homing characteristics and possess the ability to re-home to their site of origin [11]. Tumor-cell-derived extracellular vesicles (EVs) mimic the contents of their parent cell, suggesting that their therapeutically modified EVs will be biodistributed preferentially to tumors [12]. Consequently, systemically injected tumor-derived EVs will home to their original tumor tissues. Tsering with colleagues revealed that the cargo of UM-EVs play a role in premetastatic niche formation and metastasis in the liver. During metastatic progression, EVs from UM tumor cells are involved in micro-metastasis formation not detectable using current techniques. Furthermore, accumulations of progression factors promote the metastatic process [13]. 

In our previous work, we utilized the tumor-specific tropism of mesenchymal stem/stromal cells (MSCs) to develop a targeted prodrug gene therapy [14]. This therapy is mediated by MSCs transduced with a retrovirus, yielding cells engineered to express the suicide gene *yCD::UPRT*. Our retrovirus vector had the *yCD::UPRT* gene linked by an internal ribosome entry site sequence (IRES) with the neo gene, which allowed us to produce a homogenous cell population of *yCD::UPRT*-transduced cells by G418 antibiotic selection. Ultimately, the suicide *yCD::UPRT* gene is integrated into the cell DNA with a strong retrovirus promoter, the DNA provirus is expressed, and yeast cytosine deaminase catalyzed by *UPRT* converts the nontoxic prodrug 5-fluorocytosine to the effective chemotherapeutic drug 5-fluorouracil [15]. Later, we found *yCD::UPRT* transduced MSCs excreted EVs possessing mRNA of the suicide gene in their cargo [16]. Furthermore, we reported that a replicative-deficient retrovirus possessing the *yCD::UPRT* gene is able to infect several different human tumors, leading to the transduction of tumor cells with the retrovirus vector integrated into the cell DNA of the recipient tumor cells. Similar to the *yCD::UPRT*- MSCs, the tumor cells secrete EVs possessing the *yCD::UPRT* gene mRNA in their cargo [17]. Uveal melanoma, like all tumor cells, release EVs with diverse biological functions, delivering them to recipient cells. Composition and biogenesis analyses of UM EVs indicate that they induce a prometastatic microenvironment through macrophage migration inhibitory factor [18]. In addition, UM EVs have been shown to prime the liver for metastatic growth by activating stellate cells in the premetastatic niche, resulting in pro-fibrogenic properties [19]. Based on these results, as well as on our previous work, we hypothesized that the *yCD::UPRT* suicide gene-EVs secreted by primary human UM cells can attack micro-metastases and/or destroy the formation of a premetastatic niche.

In the present study, we report the transduction of several primary uveal melanoma cells and UM-associated stromal cells with the *yCD::UPRT* gene. We characterized the secreted EVs in regard to their biologic activity, tumor targeting ability, and capability to effectively kill tumor cells of different types in vitro.

## 2. Results

### 2.1. Isolation of Primary Uveal Melanoma Cells, Stromal-Associated Fibroblasts from Human Uveal Melanoma Tumors, and Their Transduction with the yCD::UPRT Suicide Gene

A schematic diagram illustrating the work flow for the isolation of the primary uveal melanoma strains and uveal melanoma-associated fibroblasts (UMAFs) is depicted in Figure 1. The expected cellular heterogeneity of the primary uveal melanoma specimens and the difficulty of introducing them to cell cultures for sustained cultivation led us to collect as many samples as we were able to obtain from patients diagnosed with UM treated by enucleation (Figure 1A). Out of the 59 tumor samples we attempted to seed into tissue culture, we were able to keep approximately 88% in cultivation. The efficiency of the tissue culture growth varied over cell passages. Seeded UM cells were infected with *yCD::UPRT*-virus in the first three days of cultivation (Figure 1B). A homogenous *yCD::UPRT*-UM cell population was obtained by selection with G418 antibiotic usually within two weeks (Figure 1C). As expected, the number of viable UM cells obtained from a patient with recurrent tumors after stereotactic radiotherapy treatment was extremely low. The morphology of primary UM cells grown in vitro was either epithelioid type (UM58), spindle shaped (UM49, UM48), or grew as spheroids when cultivated in Neurobasal medium (Figure 1D). UMAFs were isolated from primary UM114 tissue explants adhered to a plastic dish, and UMAFs were of fibroblastoid morphology (Figure 1E).

The isolated UMs and uveal melanoma UMAFs were infected with replication-defective mixed ecotropic and amphotropic envelope retrovirus containing a fused *yCD::UPRT* gene linked by *IRES* to the *neo* gene. The design of this bicistronic retrovirus vector and its activity is based on the production of an enzyme able to convert the nontoxic prodrug 5-FC to the cytotoxic 5-FU [15]. The virus-infected cells possessed an integrated provirus in melanoma cells’ DNA. Strong retrovirus promoters express both *IRES*-linked genes, thus allowing to eliminate nontransduced cells by preselected concentrations of G418 antibiotic. In addition, the presence of G418 in cultivation media has been shown to suppress the growth of stromal fibroblasts. The population purity of transduced cells can be easily checked by the addition of 5-FC into the cultivation medium, which cause the apoptotic death of all cells. We attempted to obtain UMs from each individual patient and transduce them with the suicide gene *yCD::UPRT* immediately after their isolation. We succeeded in cultivating and transducing approximately half of the UMs. The homogenous population of the *yCD::UPRT* gene-transduced cells, designated as *yCD::UPRT*-UM, were able to grow in vitro.

### 2.2. sEVs Secreted from yCD::UPRT-UMs Possess mRNA of the Prodrug Converting Gene in Their Cargo 

The evidence for the presence of the *yCD::UPRT* gene in the *yCD::UPRT*-UM transduced cells gene and equivalent nontransduced UM cells using RT-PCR visualized with agarose gel electrophoresis is presented in Figure 2. The human genome does not contain sequences relevant to the yeast cytosine deaminase that facilitates transgene PCR detection. Analysis of cell DNA from *yCD::UPRT*-UMs confirmed the presence of the integrated retroviral provirus and the absence in corresponding nontransduced UM cells (Figure 2A). The expression of the *yCD::UPRT* gene in transduced cells was proved in total cellular RNA isolates (Figure 2B). Total RNA isolated from sEVs obtained from concentrated CM by an Exoquick-TC solution precipitation, treated with DNase I, and analyzed with RT-PCR using primers for yCD amplification revealed the presence of mRNA-specific sequences (Figure 2C). 

### 2.3. Further Characterization of sEVs Secreted from yCD::UPRT-UMs 

We determined the presence of surface biomarkers using Western blot analysis in order to find the character of the excreted sEVs from *yCD::UPRT*-UMs and corresponding naïve cells, as shown in Figure 3A. The results revealed the presence of the typical markers of extracellular vesicles created through the ESCRT complex system. In addition, the integrins typical for tumor cells were identified. The quantitative presence of some markers differed exceptionally when we compared sEVs from *yCD::UPRT*-UMs with nontransduced ones. In the heterologous cell population of the primary naïve UMs, random integration of the DNA provirus in the cell DNA of the *yCD::UPRT*-UMs may influence the expression of these proteins.

### 2.4. Primary Uveal Melanoma Cells Transduced with the Suicide Gene yCD::UPRT Secrete Small Extracellular Vesicles 

The CM of the UMs was assayed for the presence of sEVs. We found that all naïve primary UMs and *yCD::UPRT*-gene-transduced cells secreted sEVs. In agreement with the reported 80 nM size of the extracellular vesicles with the round morphology excreted from the uveal melanoma cells [20], we detected similarly sized round sEVs using TEM. Both cell types produced round nanoparticles approximately 80 nM in diameter. The concentration of the *yCD::UPRT*-UM-sEVs detected with NanoSight varied between 4 and 10^10^/mL (Figure 3B). 

### 2.5. Protein Analysis of sEVs from Primary Naϊve UMs and sEVs from Corresponding yCD::UPRT-UMs

The surface protein composition of sEVs is related to the originating cell and plays a role in vesicle function [21]. The MACSPlex exosome protein marker analysis of the sEVs of the naïve primary UMs compared to the sEVs of the *yCD::UPRT*-UMs revealed surface marker profiles characterized by strong signals for the sEV associated markers: CD81, CD9, and CD63 (Figure 3C). Individual UMs differed in their expression of CD9 marker. No significant differences were found when comparing the sEVs of each individual naïve UMs with the related *yCD::UPRT* gene-transduced ones. The MACSPlex exosome capture bead data additionally indicated the presence of other markers at intermediate-to-low positive APC fluorescence intensity levels, with the highest intensity shown for CD44 and CD29, indicating the presence of other positive signals in the particular surface epitope within the exosome population (Figure 3D). The sEVs of the MP38 uveal melanoma cell line differed from the sEVs of the primary UMs in terms of CD29 marker, but the naïve MP38 sEVs and *yCD::UPRT*-MP38 sEVs revealed similar values for the MFI. Generally, there were no significant differences noted in a comparison of the naive UMs with the UMs of the *yCD::UPRT* gene transduced in regard to other markers.

### 2.6. sEVs Secreted from yCD::UPRT-UMs Alleviate Tumor Cell Growth in a Dose-Dependent Manner

The tumor-growth-inhibiting activity of CM from *yCD::UPRT* gene-transduced UMs was tested in the human melanoma cell line A375, mouse melanoma cells B16, and in an unrelated PC3 prostate cancer cell line (Figure 4). The growth inhibition of recipient tumor cells was assessed using the presence of prodrug 5-FC and in its absence using a real-time cell imaging system (IncuCyte live-cell ESSEN BioScience Inc., Ann Arbor, MI, USA). The CM from *yCD::UPRT*-UMs either stimulated or slightly inhibited the growth of tumor cells in the absence of 5-FC depending on the type of cells tested. In the presence of 5-FC, all tested tumor cells became apoptotic in a dose-dependent manner with slight differences in the growth inhibition efficacy. When 5-FC was added to the CM, the growth of the tumor cells was also inhibited in a dose-dependent manner, which varied among the different recipient cells. Generally, the best inhibition of growth was noticed with human melanoma A375 cells (Figure 4A), while the growth of mouse melanoma cells was inhibited, to a lesser extent, in a similar manner to what was found with the human prostate cancer cell line (Figure 4B,C). Even 5 µL of CM with modified sEVs caused tumor growth inhibition of approximately 60% (Figure 4B,C), while in the A375 tumor cell line, it was approximately 80% (Figure 4A) compared to a growth inhibition assay of the primary human uveal melanoma UM41 with CM from *yCD::UPRT*-gene-transduced human adipose tissue mesenchymal stromal cells (AT-MSCs).

### 2.7. Involvement of UM Stromal Cells and Their sEVs in Suicide Gene Tumor Cell Inhibition

Stromal cells forming the microenvironment play an important role in the growth of tumor cells. From primary UM114, we succeeded in isolating stromal cells—UM-associated fibroblasts (UMAFs). In order to see whether *yCD::UPRT*-UM-sEVs can inhibit or stimulate the growth of these UMAFs, we tested several *yCD::UPRT*-UM CMs for their growth ability in the presence or absence of the prodrug 5-FC. All tested *yCD::UPRT*-UM CMs partially inhibited the growth of stromal cells in the presence of 5-FC, but their growth was extremely stimulated in the absence of the prodrug (Figure 5A). Further analysis confirmed their stromal character explaining their supporting role in tumor cell proliferation. When we analyzed naïve and *yCD::UPRT* gene-transduced UMAF-114 using flow cytometry, both cells resembled mesenchymal stem cells (Figure 5B). However, cancer-associated fibroblast (CAF) markers CD140b and CD44 were found to be highly present in both naïve and *yCD::UPRT* gene-transduced cells, confirming their CAF character. In addition, this finding confirms that EVs secreted from naïve UMFs were modified to carry mRNA of the suicide gene. The absence of the CD3 marker eliminates their T-lymphocytes origin. Flow cytometry histograms of UMAF-114 and the corresponding *yCD::UPRT* gene-transduced cells are presented in Figure 5 and Appendix A Appendix A.

### 2.8. yCD::UPRT-UM-sEVs Are Responsible for the Tumor Cell-Growth-Inhibiting Activity of CM

The secretome, represented by CM from *yCD::UPRT*-UMs, is composed of biologically active factors including a number of low molecular weight proteins, such as cytokines, chemokines, growth factors, and EVs. To find out whether sEVs excreted from these cells are responsible for its cell growth-inhibiting activity, we analyzed the CM using size-exclusion chromatography on a Sephacryl S-500 column. All obtained fractions were tested for their tumor-killing activity in the presence/absence of 5-FC on A375 cells. The presence or absence of prodrug 5-FC in the assay allowed us to detect growth inhibition and/or stimulation simultaneously. Figure 6A shows that the majority of the tumor growth inhibition activity was localized to the nanoparticle fractions. 

We tested the targeting specificity of various CMs from *yCD::UPRT*-UMs tested on primary UMs of different patient origins. The CM of *yCD::UPRT*-UMs either inhibited or stimulated the growth of primary UMs individually, depending on the patient origin of the tested UMs. A comparison of the growth inhibition of the particular primary uveal melanomas or melanoma cell lines after the addition of CM from *yCD::UPRT*-UMs in the presence of 5-FC is presented in a heat map (Figure 6B). The deep green color represents the lowest growth inhibition of UMs and the pinkest the highest. Generally, almost all presented UMs and the tested melanoma cell lines A375 and MP38 were growth inhibited. The most appropriate UM cells for treatment was found to be primary uveal melanoma UM49 and the least appropriate was UM43. All tested CMs from individual *yCD::UPRT*-UMs inhibited the growth of human cell lines of hepatocyte carcinoma Hep G2, uveal melanoma MP38, human melanoma A375, and AT-MSCs. Interestingly, the cutaneous melanoma cell line A375 was found to be more sensitive to treatment than the uveal melanoma cell line MP38 (Figure 6C). 

To determine to what extent the *yCD::UPRT*-UM-sEVs were responsible for tumor growth inhibition and/or stimulation, we concentrated *yCD::UPRT*-UM-sEVs via the TFF system and tested them for cell killing activity on the cell line A375. Both products were tested for biological activity together with the original CM. A comparison was made with the number of sEVs normalized to their number in the corresponding CM (Figure 6D). The sEVs isolated from naïve primary UM41 neither inhibited nor stimulated the growth of A375 cells. As expected, analogous *yCD::UPRT*-UM41 sEVs inhibited the growth of A375 cells (Figure 6D). As can be seen in the presented data (Figure 6A,C,D), the sEVs in the absence of prodrug are not toxic to the tumor cells and, in some cases, even stimulated their growth. In the presence of the prodrug, the inhibition of growth leading to tumor cell death has always been observed. 

## 3. Discussion

Extracellular vesicles, especially sEVs secreted in large numbers from cancer cells, play an important role in metastases formation. Furthermore, an increasing amount of evidence indicates that sEVs secreted from cancer cells play a key role in the establishment of the premetastatic niche [21,22]. Generally, sEVs secreted from tumor cells are organ-tropic [23,24]. Moreover, sEVs mimic the roles performed by their cells of origin and can reach the appropriate target cells at distant sites and organs. Thanks to these features, tumor-derived sEVs are attractive targets for suicide gene modification [25]. We have previously shown in many experiments that enrichment of MSCs or tumor cells with the *yCD::UPRT* suicide gene via retrovirus construct leads to its DNA integration and expression. Consequently, the *yCD::UPRT*-transduced cells will transfer the mRNA into excreted sEVs [16,17]. Obviously, the yeast origin of the expressed gene in human cells is not compatible with cell homeostasis and is, therefore, excreted. All isolated primary uveal melanoma cells transduced with *yCD::UPRT*-retrovirus behaved similarly. They secreted EVs possessing mRNA information for the *yCD::UPRT*-suicide gene in their cargo. All tested primary *yCD::UPRT*-gene-transduced uveal melanoma cells (*yCD::UPRT*-UM) proliferated at a similar rate as their parental naïve cells. Further analysis of CM from *yCD::UPRT*-UM using size-exclusion chromatography provided evidence that sEVs in CM are responsible for their tumor cell killing activity. The presence of *yCD::UPRT* mRNA in these sEVs was probed with RT-PCR. The standard characterization of EVs, such as through using Western blot analysis and TEM, confirmed the EVs’ characteristic of nanoparticles, produced both in the transduced and naïve UMs. The concentration measurement with the NanoSight fluctuated around 4–10^10^ per ml of CM in dependence of the time–duration of the CM harvest. 

The data obtained from the bead-based multiplex flow cytometry assay confirmed the assumption that sEVs excreted from *yCD::UPRT*-UMs are the sEVs from naïve UMs enriched with mRNA of the *yCD::UPRT*-gene. The sEVs were both derived through the endosomal sorting complexes required for transport (ESCRT)-dependent sorting and exhibited integrins. The percentage value variations in the median fluorescence intensity normalized to the mean CD9/CD63/CD81 of the sEVs from individual primary UMs and sEVs excreted from the corresponding *yCD::UPRT*-UM-gene-transduced cells may reflect their individual ability to proliferate, migrate, and communicate with the environment dissemination and metastasis [26]. For the cell surface adhesion receptor CD44, we found it to be highly expressed at the same level in the sEVs from the naïve and *yCD::UPRT*-gene-transduced UMs. The CD44 receptor is highly expressed in many cancers and regulates metastasis via recruitment of CD44 to the cell surface. The same conclusion can be drawn from the observed expression data of the integrin beta-1, known as CD29, on sEVs secreted from naïve and *yCD::UPRT*-gene-transduced UMs. Integrin beta-1 activation has been shown to regulate the transition from cell dormancy to metastasis. Interestingly enough, the sEVs of MP38, both naïve and *yCD::UPRT* gene transduced, differ from the other UMs through the low expression of CD29 but again at the same level. The expression of inactive tyrosine-protein kinase transmembrane receptor ROR1 on sEVs of UMs was found to be extremely low compared to other markers, with the exception of sEVs from naive UM43. Whether the observed difference is related to the immortalized character of the UM cell line, which is likely, remains to be elucidated. This may be related to the heterogeneity of the primary UMs introduced to the growth in vitro. The *yCD::UPRT*-UM-43 cells, being selected for homogenous *yCD::UPRT*-gene-transduced cells morphologically, resemble stromal fibroblastoid cells. Generally, all of these cytometric data show that the transduction of primary UMs with *yCD::UPRT* retroviral construct is not leading to the dramatic changes in the immunologically detected sEVs markers, despite the random integration of retrovirus into the cell DNA, which could cause changes in the transcription of neighboring genes near to the integrated provirus and general heterogeneity of tumor-derived extracellular vesicles. 

The host range of the *yCD::UPRT*-UM-sEVs in the tumor cell growth inhibition in the presence of prodrug was found not to be restricted to UMs only. Furthermore, the specificity of the sEVs from the individual primary UM and from them derived *yCD::UPRT*-UM-sEVs were found not to show differences between patients from whom the cells were obtained. Generally, highly proliferating cells that are metabolically more active excrete higher numbers of EVs. However, some UM cell lines, primary UM cells, and *yCD::UPRT*-gene-transduced cells proliferate very slowly. In addition, the proliferation of primary UMs drops with an increasing number of passages. 

The proliferating activity of individual UMs is in good agreement with the histopathologic findings of cyclin D1 expression in individual uveal melanoma tumor tissue. This could be the reason why some biomarkers in UM37 and *yCD::UPRT*-UM43 sEVs tested negative in the WB analysis. Cyclin D1 was found to be highly overexpressed in UM49, UM55, and UM41 cells, while in UM37 and UM43 cells, the cyclin D1 expression was at the limit of detection. The heterogeneity of the sEVs populations is probably responsible for the broad host range of the *yCD::UPRT*-UM-sEVs tested in vitro on the human and mouse tumor cell lines of various types. 

We hypothesized that sEVs excreted from UM modified with suicide gene messages may be targeted to liver metastases and/or prevent the formation of premetastatic niches. Human hepatocyte carcinoma Hep G2 was found to be sensitive to *yCD::UPRT*-UM-sEVs present in CM in our in vitro experiments. We speculate that *yCD::UPRT*-gene-modified sEVs secreted from primary UMs being targeted to liver may act as a preventive vaccine against liver UM metastases. The *yCD::UPRT*-UM-EVs together with the prodrug systematically injected could prevent metastases formation by the inhibition of premetastatic niche creation. This assumption is supported by the evidence that *yCD::UPRT*-MSC-EVs together with the prodrug was not toxic in rats, we found, when injected systematically to treat intracerebral rat glioblastoma with curative outcome [27]. Several recent reports support this notion. For example, genetically modified tumors releasing sEVs were found to possess antitumor potential [28]. Recently, engineered tumor exosomes were found to act as a dendritic cell (DC)-primed vaccine, which boosted antitumor immunity in breast cancer [29]. Furthermore, MSC exosomes modified with the *yCD::UPRT* gene used for the experimental treatment of glioblastoma-carrying rats prevented the recurrence of glioblastoma in cured rats [27]. The prevention of UM metastases formation via the inhibition of premetastatic niche creation by *yCD::UPRT*-UM-sEVs seems to be possible. In vivo experiments with primary UM using patient-derived xenografts in immunocompromised mice will be the next translational approach.

EVs secreted from cells, as a natural cell product, have many advantages over conventional nanocarriers [30]. They have low immunogenicity, good biocompatibility, can penetrate the natural blood–brain barrier, and have the capacity for gene delivery. They can be loaded with anticancer drugs; consequently, EVs are absorbed in much greater proportions than other nanocarrier vesicles, such as liposomes or polystyrene NPs [31]. Cell-secreted sEVs have a distinct cell tropism and homing selectivity that can be used to target them into certain tissues or organs [32]. For instance, sEVs derived from dental pulp MSCs with gemcitabine as a cargo have an inhibitory effect on the growth of pancreatic carcinoma cell lines in vitro [33].

## 4. Materials and Methods

### 4.1. Establishment of Cell Cultures from Human Uveal Melanoma 

Uveal melanoma cells were obtained from eye globes with uveal melanoma (n = 59) enucleated at the Department of Ophthalmology (Medical Faculty of the Comenius University, Bratislava, Slovakia) between 2019 and 2020. All of the enucleated eye globes with uveal melanoma were histopathologically verified. Donors were informed about the nature of the study, and they provided their written informed consent. The study was approved by the Ruzinov Hospital Bratislava Ethics Committee on 12 December 2018 (number: EK/250/2018). Each UM tumor was processed within two hours after enucleation, disintegrated by mincing, and followed by enzymatic digestion of the tumor fragments with a 0.25% trypsin-EDTA solution (Sigma Aldrich, St. Louis, MO, USA) at 37 °C for 15 min. The cells were cultured as either monolayers or multicellular spheroids at 37 °C in a humidified atmosphere of 95% air and 5% CO_2_ with MEMα medium (Gibco, Thermofisher Scientific, Waltham, MA, USA) supplemented with 10% fetal bovine serum (FBS) (Biosera, Nuaille, France). Serial passages were performed by trypsinization (0.25% trypsin-EDTA solution) of subconfluent monolayers. The cells were seeded in a tissue culture dish (Corning Life Sciences, Corning, NY, USA) at a density of 5 × 10^4^ cells/cm^2^, and the culture medium was replaced every 2–3 days. 

### 4.2. Isolation of Uveal Melanoma-Associated Fibroblasts (UMAFs) 

In order to isolate the stromal cells, uveal melanoma tissue fragments were adhered to the plastic tissue culture dish and surrounded with a complete culture medium of low-glucose DMEM (1 g/L) supplemented with 5% human platelet lysate (HPL). The next day, the fragments were overlaid with medium and further cultivated at 37 °C in a humidified atmosphere with 5% CO_2_. Two–three days later, the stromal cells—UMAFs—started to migrate from the adhered UM tissue explants. 

### 4.3. Tumor Cell Lines’ Maintenance 

The lines used in this study were the human melanoma cell line A375, mouse melanoma B16-F0, and human prostate carcinoma cell line PC3. All were maintained in high-glucose DMEM with 5% FBS. The cells were authenticated using STR profiling. The uveal melanoma cell line MP38 and human hepatocyte carcinoma Hep G2 cell line were purchased from ATCC (American Type Culture Collection, Manassas, VA, USA) and maintained in high-glucose DMEM supplemented with 10% FBS. All cell lines were routinely tested for mycoplasma infection using PCR, and all experiments were performed using mycoplasma-free cells.

### 4.4. Preparation of Replication-Deficient yCD::UPRT Suicide Gene Retrovirus

The procedure for the preparation of cells producing *yCD::UPRT*-retrovirus was performed as previously described [15]. Briefly, we used a bicistronic retroviral vector containing the fusion gene yeast cytosine deaminase uracilphosphoribosyl transferase (*yCD::UPRT*) separated by *IRES* sequences from the *neo* gene allowing for the selection of a pure population of transfected cells with Geneticin (G418) antibiotics. The cells that produced the virus with mixed eco- and ampho- glycoprotein virus envelopes were generated with four ping-pong rounds of infections of helper cell lines GP+E-86 and GP+envAm12.

### 4.5. Transduction of UM Cells with yCD::UPRT-Retrovirus

In order to obtain a transduced population of primary UM, the semi-confluent cell culture was infected with *yCD::UPRT*-retrovirus supplemented with protamine sulphate (5 μg/mL). A homogenous population of *yCD::UPRT*-gene-transduced UM cells was obtained by cultivation in selection media with G418. We used a pretested concentration of G418 for selection in every primary UM cell culture using a range over 0.4 to 1.0 mg/mL.

### 4.6. Cell Growth Assessment using the IncuCyte Live Cell Monitoring System

The growth of the UM cell cultures and various cancerous cell lines under the designed treatment was monitored using the IncuCyte ZOOM Kinetic Imaging system (Essen BioScience, Royston, UK). Cell viability data obtained from the IncuCyte ZOOM system were compared with data obtained with the standard MTT test—CellTiter 96 Aqueous One Solution Cell Proliferation Assay (Promega, Madison, WI, USA).

### 4.7. Conditioned Media and Extracellular Vesicles Isolation 

Conditioned medium (CM) was used as the source of the sEVs in the experiments where the biological activity of the *yCD::UPRT*-UM-sEVs was evaluated and for the particle size estimation. The isolation of the sEVs was performed for the downstream analyses, such as Western blotting and PCR. The CM was harvested from the semi-confluent UM cell cultures, washed three times with phosphate buffered saline (PBS) to remove any debris, and then cultured in FBS-free MEMα for 24 or 48 h. The 24 h CM was collected every second day, and the 48 h CM was collected every third day, alternating with the growth culture fluid exchange. The usual number of cells was 2 × 10^6^ cells in a 100 mm Petri dish. The harvested CM was centrifuged at 800× *g* for 5 min to remove any possible residual cells and cell debris and then filtered through a 0.22 μm syringe filter to remove microvesicles. The sEVs were isolated from the CM using size-exclusion chromatography, Amicon^®^ Ultra-15 ultrafiltration, TFF-Easy—tangential flow filtration EV concentrator 1 (HansaBioMed Life Sciences Ltd., Tallinn, Estonia), or ExoQuick TC (System Biosciences, Mountain View, CA, USA). Isolation of RNA from the sEVs was performed with a SeraMir Exosome RNA Column Purification Kit (System Biosciences, Mountain View, CA, USA). The sEVs for the Western blot analysis were concentrated 20–40 times using an Amicon^®^ Ultra-15 Centrifugal Filter Unit 100 kDA MWCO (Merck KGaA, Darmstadt, Germany) and precipitated with ExoQuick TC (System Biosciences, Mountain View, CA, USA).

### 4.8. Separation of the sEVs from Conditioned Media with Size-Exclusion Chromatography

The CM from the *yCD::UPRT*-transduced uveal melanoma cells was separated in a PD-10 column (GE Healthcare, 105 mm × 15 mm) filled with Sephacryl S-500 (GE Healthcare, Chicago, IL, USA), pre-equilibrated with PBS, and stored at 4 °C in 20% ethanol. Before use, the column was equilibrated to room temperature and washed with PBS (two void volumes of the column) using a peristaltic pump P-1 (Pharmacia Lkb, Uppsala, Sweden). Subsequently, 1 mL of filtered CM (0.22 µm PES filter) was applied in the column, and 22 fractions (0.5 mL each) were collected. All fractions were tested for tumor cell proliferation (in the absence of 5-FC) and growth inhibition (in the presence of 5-FC) on A375 cells monitored with an IncuCyte S3 Live-Cell Analysis System.

### 4.9. Evaluation of sEVs’ Concentration and Size Distribution 

The assessment of the particle size, quantity, and distribution was performed using a NanoSight NS500 instrument (Malvern Instruments Ltd., Great Malvern, UK) equipped with a 405 nm laser and sCMOS trigger camera. Data were analyzed using Nanoparticle Tracking Analysis (NTA) 2.3 software. 

### 4.10. Cell DNA Isolation and PCR Amplification 

DNA extraction from cell pellets was performed using a QIAmp DNA Mini kit (Qiagen, Venlo, The Netherlands) following the manufacturer’s instructions. A NanoDrop^®^ ND-1000 spectrophotometer (Thermo Fisher Scientific, Wilmington, DE, USA) was used to control for extracted DNA quantity and quality. Standard PCR using 50 ng of genomic DNA was then performed with the same sets of primers and chemistry as qPCR employing a Bio-Rad T100 thermal cycler (Bio-Rad, Hercules, CA, USA). The products were visualized by agarose gel electrophoresis (2%) using a 50 bp DNA ladder.

### 4.11. Cell RNA Isolation and RT-PCR

Total RNA was extracted from pellets of *yCD::UPRT*-UM cells and corresponding control cells with RNeasy Mini Kit (Qiagen, Venlo, The Netherlands). A NanoDrop^®^ ND-1000 spectrophotometer (Thermo Fisher Scientific, Wilmington, DE, USA) was used to assess the RNA quality and quantity. Reverse transcription was performed using a RevertAid First Strand cDNA Synthesis Kit (Thermo Scientific, Loughborough, UK). Real-time PCR (RT-PCR) was performed with identical primers while following the same protocol used for exoRNA. The GAPDH gene was used as an amplification control.

### 4.12. Exosome RNA Isolation 

The sEVs from the cultivation media were isolated using the Complete SeraMir Exosome RNA Amplification kit (System Biosciences, Mountain View, CA, USA) containing Exoquick-TC (System Biosciences, Mountain View, CA, USA), using the protocol recommended by the supplier. Briefly, CM was centrifuged at 3000× *g* for 15 min to remove cells and cell debris. Then, 5 mL of supernatant was mixed with 1 mL of Exoquick solution, mixed, and then incubated overnight at 4 °C. Subsequently, the tubes were centrifuged at 16,000× *g* for 2 min, and the sEVs pellet was reconstituted in 350 μL of lysis buffer. After vortexing, followed by incubation for 5 min at room temperature to allow for complete lysis, 200 μL of 100% ethanol was added, and 600 μL of the mixture was filtered through the spin column at 16,000× *g* for 1 min. Two wash steps were then performed with 400 μL of wash buffer, followed by centrifugation at 16,000× *g* for 1 min, which was then followed by the elution of total RNA into 30 μL of elution buffer. The quality of the eluted exoRNA was then evaluated using the Agilent RNA 6000 Nano Kit (Agilent Technologies, Santa Clara, CA, USA).

### 4.13. cDNA Synthesis from Total RNA

Five microliters of the eluate was then used for reverse transcription using the Complete SeraMir Exosome RNA Amplification kit (System Biosciences, Mountain View, CA, USA). For the poly A reaction, exoRNA eluted from the spin column (5 µL), 5× polyA buffer (2 µL), 25 mM MnCl_2_ (1 µL), 5 mM ATP (1.5 µL), and polyA polymerase (0.5 µL) were incubated for 30 min in 37 °C. Then, 0.5 µL of SeraMir Adaptor Oligo was added, and the mix was incubated for 5 min at 60 °C. After subsequent incubation at room temperature for 2 min, the RT reaction was performed in a 20 µL volume and consisted of polyA exoRNA from the previous step (10 µL), 5× RT master mix (4 µL), 5’ SeraMir Switch Oligo (1 µL), reverse transcriptase (1 µL), and RNA-se-free water (4 µL). Incubation in a BioRad T100 thermal cycler (Bio-Rad, Hercules, CA, USA) at 42 °C for 30 min was followed by 95 °C for 10 min to terminate the reaction.

### 4.14. Real-Time PCR Profiling Using exo-cDNA

Real-time PCR (RT-PCR) was then performed using 20 µL of the cDNA generated in the previous step, using 10 μL of 2× GoTaq^®^ qPCR Master Mix (Promega, Madison, WI, USA), 1 μL (0.5 μM) forward primer, 1 μL (0.5 μM) reverse primer, 7 μL ultrapure DNase/RNase-free water, and 1 μL of cDNA. Primer sequences were as published previously [17]. 

*yCD::UPRT* forward: 5’-ATGGACATTGCCTATGAGGA-3’; 

*yCD::UPRT* reverse: 5’-TTCTCCAGGGTGCTGATCTC-3’ (product size 167 bp); 

*GAPDH* forward: 5’-GAAGGTGAAGGTCGGAGTC-3’; 

GAPDH reverse: 5’-GAAGATGGTGATGGGATTTC-3’ (product size 226 bp). 

All samples were analyzed in triplicate on a Bio-Rad CFX96 real-time PCR detection system (Bio-Rad, Hercules, CA, USA). The following cycling conditions were applied: 95 °C for 10 min, 40 cycles at 95 °C for 15 s, 60 °C for 30 s, and 72 °C for 30 s. Given the missing *yCD::UPRT* transcripts in negative controls, the presence of specific transcripts was visualized using agarose gel electrophoresis.

### 4.15. Western Blot Analysis 

The sEVs were lysed in RIPA buffer with a protease inhibitor cocktail (Complete ULTRA, Roche, Switzerland), a phosphatase (PhosSTOP, Roche, Switzerland), and sonicated. The proteins (30 μg per well) were loaded onto 8% or 10% SDS polyacrylamide gels and subjected to electrophoresis. Separated proteins were transferred to a 0.45 µm Nitrocellulose Transfer Membrane (Thermo Fisher Scientific) using the Mini Trans-Blot Cell Module (Bio-Rad Laboratories, Inc., Hercules, CA, USA). Membranes were incubated overnight at 4 °C with the appropriate primary antibodies (CD9 #10626D, CD63 #10628D, and CD81 #10630D purchased from Invitrogen, Integrin α5/CD49e #610633 from BD Biosciences, and Integrin β1 Antibody #4706S from CST). The applied dilutions are shown in Appendix A Appendix A. A primary monoclonal antibody against β-actin (# A1978 Sigma-Aldrich, St-Louis, MO, USA) served as a loading control. The specific binding of the antibodies was detected with the appropriate peroxidase-conjugated Alexa Fluor 680 secondary antibody and visualized using an LI-COR^®^ instrument (LI-COR^®,^ Lincoln, NE, USA).

### 4.16. Bead-Based Multiplex Flow Cytometry Assay 

The sEVs isolated from the primary UMs and sEVs from the corresponding *yCD::UPRT*-UMs were subjected to a surface-marker characterization using a flow cytometry bead-based multiplex analysis (MACSPlex Exosome Kit, human, Miltenyi Biotec. 130-108-813). The samples were processed according to the manufacturer’s protocol. Briefly, sEVs were isolated via CM concentration using a TFF-Easy EV concentrator 1 (TFF). With TFF, a filter unit separates the CM into retentate containing sEVs and flow-through permeate. One-third of the 50-fold concentrated CM sEVs were mixed with the manufacturer’s buffer to a final volume of 120 µL, followed by the addition of 15 µL of MACSPlex Exosome Capture Beads. After, the samples were incubated overnight at room temperature, protected from the light, and on rotation—450 rpm. Then, 500 µL of the MACSPlex buffer was added, and the samples were centrifuged at 3000× *g* for 5 min. The same volume of supernatant was discarded, and 15 µL of the detection antibody cocktail—5 µL of each MACSPlex exosome detection reagent: CD9, CD63, and CD81—was added to each sample. The samples were incubated for 1 hour at room temperature, protected from the light, and on rotation—450 rpm. Later, 500 µL of buffer was added to each sample, centrifuged at 3000× *g* for 5 min, followed by discarding of the same volume of supernatant, and washed again with 500 μL of buffer. The samples were incubated for 15 min at room temperature on rotation, protected from light, and then centrifuged for 5 min at 3000× *g*. Subsequently, 500 µL of the supernatant was removed, and approximately 150 µL of each sample was re-suspended and used for the analysis. The flow cytometry analysis was performed using the MACSQuant^®^ Analyzer 10 (Miltenyi Biotec), and results were processed with MACSQuant Analyzer 10 software (Miltenyi Biotec). The 39 single bead populations were gated to determine the APC signal intensity on each bead population, and the median fluorescence intensity (MFI) for each capture bead was measured. For each population, the background was corrected by subtracting the respective MFI values from the non-EVs controls that were treated exactly like the EV samples and, furthermore, normalized to the mean of the APC-MFI detected for the CD9, CD63, and CD81 Exosome Capture Beads (mean CD9, CD63, and CD81 = 100%). Only positive markers are shown in the graphics. 

### 4.17. Statistical Analysis

Unless noted otherwise, all experiments were repeated at least three times to enable statistical analysis, and all results were similar among replicates. Data from the MACSPlex analysis were statistically evaluated using the one-way ANOVA nonparametric test (Friedman test). For multiple comparisons, Dunn’s test was used. The statistical analysis was carried out using GraphPad Prism 9 software (GraphPadPrism Software, San Diego, CA, USA).

## 5. Conclusions

Extracellular vesicles (EVs) of human uveal melanoma (UM) play roles in processes leading to the development of premetastatic niche and metastases specifically located in the liver. We isolated primary uveal melanoma cells and modified secreted EVs to carry mRNA of a suicide gene (yeast cytosine deaminase::uracil phosphoribosyl transferase (*yCD::UPRT*). *yCD::UPRT*-UM-EVs cause tumor cell death, converting a prodrug (5-fluorocytosine) into a cytotoxic cancer drug 5-fluorouracil intracellularly. The *yCD::UPRT*-UM-sEVs hold the therapeutic potential to become a therapeutic drug for UM. However, the use of *yCD::UPRT*-UM-sEVs must first be tested in animal preclinical studies.

## Figures and Tables

**Figure 1 ijms-24-12957-f001:**
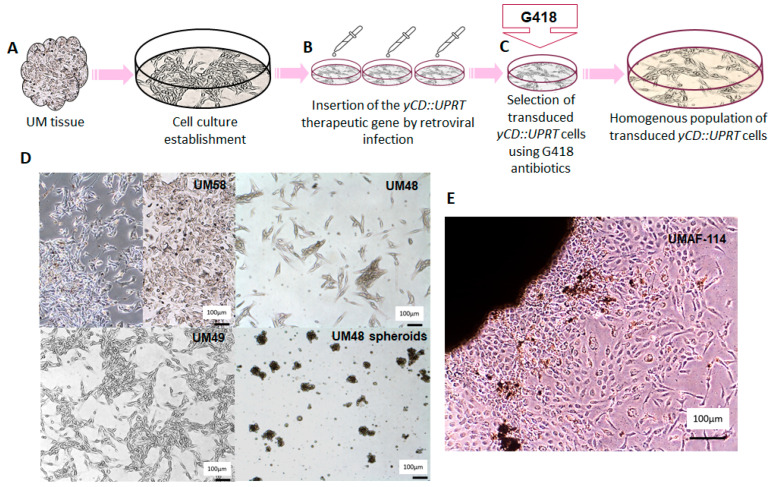
Schematic work flow of UM cell isolation and subsequent transduction with the *yCD::UPRT* gene: (**A**) outline of primary UM cell isolation; (**B**) seeded uveal melanoma cells were infected with *yCD::UPRT*-virus in the first three days of cultivation; (**C**) homogeneous *yCD::UPRT*-UM cell population selected by medium containing G418; (**D**) morphology of UM cells grown in vitro, either epithelioid type (UM58), spindle shaped (UM49, UM48), or spheroid when cultivated in Neurobasal medium; (**E**) isolation of UMAFs by stromal cells realized from an explant of a UM tumor tissue specimen.

**Figure 2 ijms-24-12957-f002:**
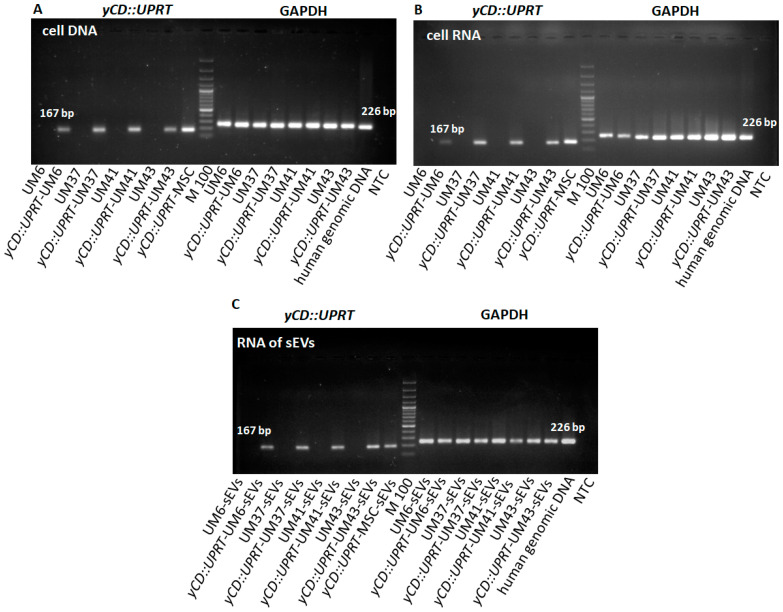
RT-PCR analysis of *yCD::UPRT* sequences in transduced and naïve cells and corresponding sEVs: (**A**) presence of DNA-integrated *yCD::UPRT* gene in UM cells; (**B**) detection of mRNA expression in transduced *yCD::UPRT*-UMs; (**C**) detection of mRNA of *yCD::UPRT* in sEVs.

**Figure 3 ijms-24-12957-f003:**
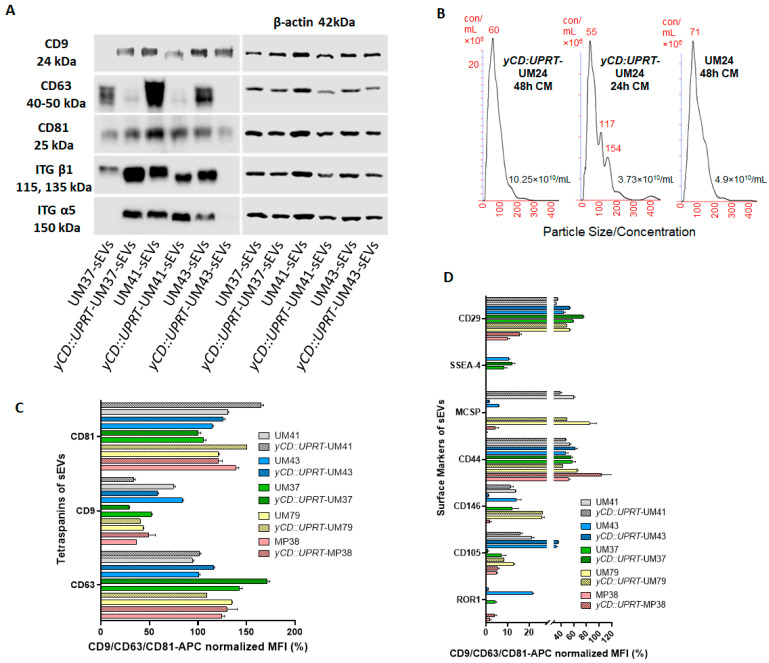
Further characterization of the sEVs secreted from the *yCD::UPRT*-UMs: (**A**) Western blot analysis of specific surface exosomal biomarkers and integrins was evaluated using a chemiluminescence-based method from *yCD::UPRT* gene-transduced and equivalent nontransduced UM sEVs. (**B**) graphs represent the size and concentration of sEVs obtained from conditioned media of transduced and nontransduced cells after 24 and 48 h. Exosome size analysis and concentration determination was performed using a NanoSight NS500 Instrument (Malvern Instruments, Malvern, UK). (**C**) Levels comparison of the characteristic sEVs tetraspanins—sEVs of the primary UMs with sEVs of the corresponding *yCD::UPRT*-UMs and sEVs of the MP38 human uveal melanoma cell line with equivalent sEVs of *yCD::UPRT*-MP38 were compared. (**D**) Flow cytometric MACSPlex analysis of additional surface protein markers. UM-sEVs, corresponding *yCD::UPRT*-UM-sEVs, and sEVs of the MP38 human uveal melanoma cell line with analogous sEVs of *yCD::UPRT*-MP38 were compared. The y-axis represents the protein marker profile, whereas the *x*-axis shows the normalized APC-MFI. The median APC signal intensity of each specific population of single beads was normalized to the average of the anti-CD9, anti-CD63, and anti-CD81 beads.

**Figure 4 ijms-24-12957-f004:**
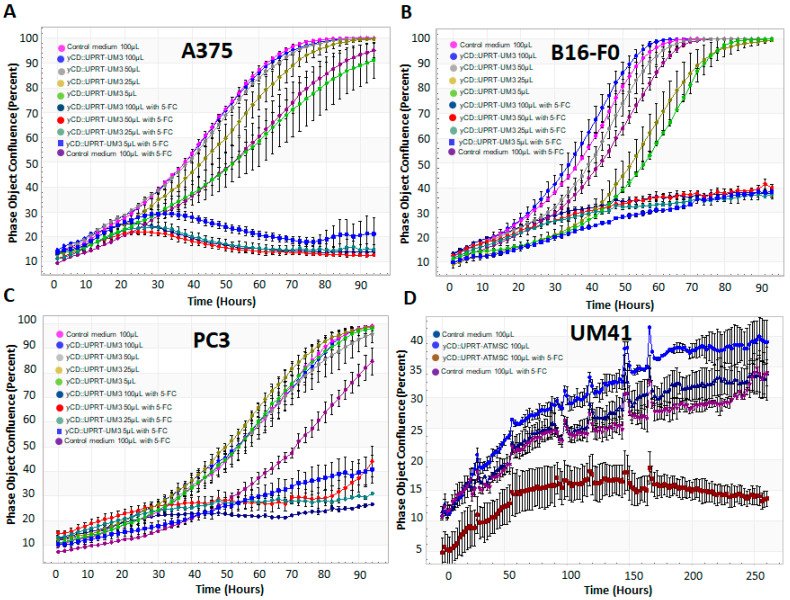
Assessment of the growth inhibition efficiency of CM from *yCD::UPRT*-UM3 tested in three different cancer cell lines compared to the growth inhibition efficiency of *yCD::UPRT*-AT-MSC-CM in primary UM41 cells. Tumor cell growth inhibition was tested in the presence and absence of the prodrug 5-FC. Tumor cell growth was monitored in 96-well plates using a real-time live cell scanning IncuCyte system: (**A**) growth inhibition assay of the human cutaneous melanoma cell line A375; (**B**) growth inhibition assay of the mouse melanoma cell line B16-F0; (**C**) growth inhibition assay of the human prostate cancer cell line PC3; (**D**) growth inhibition assay of the primary human uveal melanoma UM41 with *yCD::UPRT*-AT-MSC-CM.

**Figure 5 ijms-24-12957-f005:**
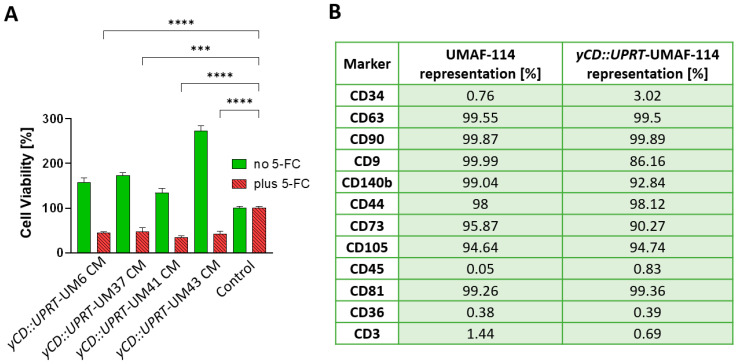
Tumor-cell-killing activity of uveal melanoma-associated fibroblasts and their protein surface marker characterization. Cell viability (%) was evaluated as a percentage relative to the untreated control cells. (**A**) Efficacy of CM from *yCD::UPRT*-UM6, *yCD::UPRT*-UM37, *yCD::UPRT*-UM41, and *yCD::UPRT*-UM43 in the growth inhibition and/or stimulation of UMAF-114 cells. Statistical analysis was performed by comparing treated samples plus 5-FC to an untreated control sample plus 5-FC using one-way ANOVA with Welch’s correction. Values of *p* < 0.0001 (****) and *p* < 0.0002 (***). (**B**) Flow cytometric analysis of UMAF-114 cells and corresponding *yCD::UPRT*-UMAF-114.

**Figure 6 ijms-24-12957-f006:**
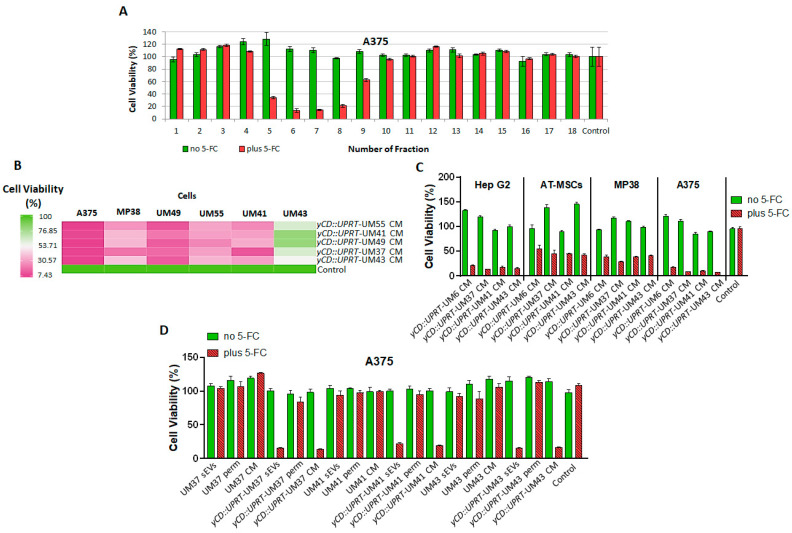
Evidence for the biological activities of sEVs excreted from UMs and *yCD::UPRT*-UMs. Cell viability (%) was evaluated as a percentage relative to the untreated control cells. (**A**) Size-exclusion chromatography of CM from *yCD::UPRT*-UM4 cells on Sephacryl S-500 columns. The tumor-cell-killing activity against A375 cells was assessed in all fractions with and without the presence of 5-FC using the MTT test. (**B**) Heat map presentation of the tumor cell-growth-inhibiting effect of CM from *yCD::UPRT*-UM55, *yCD::UPRT*-UM41, *yCD::UPRT*-UM49, *yCD::UPRT*-UM37, and *yCD::UPRT*-UM43, all of which contained sEVs possessing mRNA of *yCD::UPRT*-gene tested in the presence of 5-FC in the cutaneous melanoma cell lines A375, uveal melanoma cell line MP38, and primary uveal melanoma UM49, UM55, UM41, and UM43 cells. (**C**) Growth inhibition of the tumor cell lines Hep G2, MP38, A375, and adipose tissue-derived MSCs with sEVs present in 100 µL of CM from *yCD::UPRT*-UM6, *yCD::UPRT*-UM37, *yCD::UPRT*-UM41, and *yCD::UPRT*-UM43 cells with or without 5-FC. The statistical analysis of the data was performed by comparing treated samples plus 5-FC to an untreated control sample plus 5-FC using one-way ANOVA with Welch’s correction. All analyzed data presented are statistically significant with a value of *p* < 0.0001, except in the case of the sample *yCD::UPRT-UM6* CM plus 5-FC tested in human adipose tissue-derived MSCs, with a value of *p* < 0.0002. (**D**) The biological activity of the retentate (sEVs), permeate (perm), and CM from primary UMs and corresponding *yCD::UPRT*-UMs isolated with TFF was tested in A375 cells. The statistical analysis of the data was performed by comparing *yCD::UPRT*-UM-sEVs and *yCD::UPRT*-UM-CM-treated samples plus 5-FC to an untreated control sample plus 5-FC using one-way ANOVA with Welch’s correction. All analyzed data were statistically significant with value of *p* < 0.0001.

## Data Availability

The Data Availability Statement Material mentioned in the article is available upon reasonable request.

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
