# Peer review of "Suicide-Gene-Modified Extracellular Vesicles of Human Primary Uveal Melanoma in Future Therapies"

_ijms, 2023, doi:10.3390/ijms241612957_

Round 1
Reviewer 1 Report
7 August 2023
Ms. Ref. No.: ijms-2556048
Journal: International Journal of Molecular Sciences.
Title: Anti-metastatic potential of suicide gene modified extracellular vesicles of human primary uveal melanoma
Comments:
Thank you for your efforts in writing this article on a very pertinent topic. Moreover, I found the article to be informative and with the potential for further research on this topic in future.
I have some observations where mentioned in the following paragraphs that will be useful for its improvement:
1- To begin with, there are only a few references available. It seems that can adding further suitable references.
2- Please review the manuscript for any grammar errors.
3- What was the purpose of using the PC3 cell line in relation to the connection between uveal melanoma and liver metastasis?
4- The methods section could be more succinct and some information did not need to be included.
5- The figures were explained both in the text and through captions, allowing for easy reference to them in the text.

It can be Re-edited again.
Reviewer 2 Report
In the manuscript entitled „Anti-metastatic potential of suicide gene modified extracellular vesicles of human primary uveal melanoma”, Jakubechova and co-authors transduced primary uveal melanoma cells or fibroblasts derived from human tumour tissue with a suicide gene. The authors showed that RNA for the suicide gene is present in small extracellular vesicles (sEVs) produced by the transduced cells. Furthermore, conditioned medium or sEVs from these cells can apparently transfer the suicide gene to other tumour cells, as the viability of the latter was reduced when they were treated with 5-FC, a nontoxic prodrug that is converted to the cytotoxic drug 5-FU by the suicide gene.
This manuscript follows up on substantial previous work by the authors, in which they have expressed the same suicide gene either in mesenchymal stem cells (MSCs) or in tumour cells and have shown that the MSCs or conditioned medium derived from MSCs and tumour cells can confer suicidal abilities to other tumour cells exposed to 5-FC (refs 14-17, 28 from the manuscript). In particular, the publication by Altanerova et al, Int J Cancer 2021, transduced the suicide gene in various types of tumour cells and showed that conditioned medium from these cells could inhibit the growth of several other types of tumour cells when treated with 5-FC. The current study essentially replicates these findings using an additional type of tumour, uveal melanoma. The conceptual novelty of the manuscript is thus limited, but the results can be seen as a useful confirmation of the previous work and could be of specific interest for those working with uveal melanoma.
Overall, the experiments appear properly controlled. However, some of the central claims of the paper are not supported by the data. In addition, there are specific deficiencies in data presentation, as detailed below:
1) The title claims that suicide gene modified extracellular vesicles have anti-metastatic potential in uveal melanoma. This is completely misleading and has zero basis in the manuscript’s data. The authors have done no experiments related to metastasis at all. In fact, they have not even examined primary tumour growth, as all of the experiments are performed in cell culture only. Therefore, any claims or suggestions linking this study to metastasis (or premetastatic niches) are completely inappropriate. The authors should either perform experiments to assess metastasis in vivo or they must remove any claims or suggestions related to effects on metastasis – not only from the title, but also from various other parts of the manuscript, including the abstract, introduction, discussion and conclusion.
2) Another way in which the title, abstract and other parts of the text are misleading, is that in most cell viability/growth experiments the authors did not use extracellular vesicles, but conditioned medium, which contains many other components. The only functional experiment that used extracellular vesicles is the one shown in Fig. 6D and in this one the viability of cutaneous melanoma cells was assessed, not uveal melanoma. However, the authors claim that the “yCD::UPRT-UM-sEVs inhibited the growth of the human cutaneous melanoma cell line A375, uveal melanoma cell line MP38, as well as other primary UMs to various extents in vitro” (Abstract) and “Human hepatocyte carcinoma Hep G2 was found to be sensitive to yCD::UPRT-UM sEVs in our in vitro experiments” (Discussion). The authors need to provide direct evidence for these claims, which is currently missing.
3) The authors claim that the sEVs containing the suicide gene can be used therapeutically. This raises the question whether delivery of the suicide gene through sEVs provides any advantages compared to other vehicles, such as liposomes, lipid nanoparticles, polymeric nanoparticles, cationic nanoemulsions etc. A simple question that could be addressed first would be how the efficiency of suicide gene delivery by sEVs (and the ensuing sensitivity to 5-FC) compares to standard transfection techniques in vitro. The authors should at least discuss this issue.
4) The results of viability assays in Fig. 5 and Fig. 6 are presented in “%”, but it is not made explicitly clear what the percentage relates to. I assume that viability in control, unconditioned medium was set as 100%, but the authors should specify this.
5) On p. 12 it is stated: “All analyzed data presented in Figure 6D were statistically significant with value p<0.0001 and p<0.0002 except in case of the sample yCD::UPRT-UM6 CM plus 5-FC tested in human adipose tissue derived MSCs.” First, what was statistically significant compared to what, using what tests? Second, it appears that this description might refer to Fig. 6C, not 6D.
Related to the above, the legend to Fig. 6 states ”Value p<0.0001(****), p<0.0002(***)”, but there are no “stars” are in the figure itself.
6) Page 9 contains the following statements: “On the other hand, sEVs of MP38 uveal melanoma cell line differed from primary UMs in these markers, but again the sEVs and yCD::UPRT-MP38 revealed similar value of MFI. In these markers sEVs from melanoma cell line MP38 differed significantly from primary UMs both naïve and gene transduced.” After struggling for a while to understand what the authors mean, I was still left unsure. These statements must be revised for clarity.
7) A supplementary Table 1 is mentioned in the Methods, but I did not find such a table in the submitted files.
8) In Fig. 3C, the order of the bars is the reverse of that in the legend, which makes it more difficult to assess the graph.
The manuscript is generally comprehensible, but the reader often has to struggle with poor syntax, clunky sentence structure and inappropriate word choice, which substantially impairs the readability of the text. The authors are advised to have their manuscript revised either by a manuscript editing service or at least by some of the automatic text optimisation tools available.
